

# Parents' evaluation of support in Australian hospitals following stillbirth

Melanie L. Basile and Einar B. Thorsteinsson

School of Behavioural, Cognitive and Social Sciences, University of New England, Armidale, NSW, Australia

## ABSTRACT

The present study evaluated the level of support and satisfaction among parents of stillborn babies in Australian hospitals. One-hundred and eighty-nine mothers and fathers completed an online survey designed by the researcher based on the guidelines designed by the Perinatal Society of Australia and New Zealand. Support was inconsistent with guidelines implemented on average only 55% of the time. Areas of support regarding creating memories, birth options and autopsy were most problematic. A significant positive correlation was found between support and satisfaction and there is indication that there has been some increase in support and satisfaction over time. There has been a significant increase in both support and satisfaction since the release of the guidelines in 2009. Creating memories was regarded by parents as the most influential to their grief. It is recommended that health professionals review guidelines and seek feedback from parents as to how they can improve the support they provide.

## INTRODUCTION

### Scope of the problem

A stillbirth, as defined in Australia, is the death of a baby before or during birth at 20 weeks or more gestation, or with a birth weight of at least 400 grams (*Li et al., 2013*). In 2011, there were 2,200 stillbirths in Australia providing a rate of 7.4 stillbirths per 1,000 total births (*Li et al., 2013*).

Stillbirth is a unique loss in that it encompasses the loss of a person, loss of parenthood, and the loss of future hopes and dreams (*Boyle et al., 1996*; *Fetus and Newborn Committee, 2001*; *Kowalski, 1983*; *Robinson, Baker & Nackerud, 1999*). This devastating loss is compounded by the lack of acknowledgement, validation, and support in the community (*Bennett et al., 2005*; *Kowalski, 1983*; *McGreal, Evans & Burrows, 1997*). The grief may extend into future pregnancies and can last from months to years (*Hutti, 2005*). Mothers of stillborn babies are up to three times more likely to develop anxiety and depression, and suffer significantly higher levels of psychological distress compared with mothers of living infants (*Boyle et al., 1996*; *Rådestad et al., 1996*). Feelings of shock, guilt, anger, anxiety, emptiness, loneliness, and helplessness are often expressed as families are ill-prepared to deal with the intensity of their grief (*Boyle et al., 1996*; *Callan & Murray,*

Corresponding author
Einar B. Thorsteinsson,
ethorste@une.edu.au

*1989*; *Flenady et al., 2014*). Undermining the damage following a stillbirth leaves bereaved families to grieve in isolation and silence (*Scott, 2011*).

The Perinatal Society of Australia and New Zealand (PSANZ) have developed evidence-based guidelines to assist health professionals in providing relevant support for bereaved parents (*Flenady et al., 2009*). In an attempt to ensure that support in Australia is consistent and meeting the needs of bereaved parents, a thorough evaluation must take place. The current study aims to explore levels of perceived support and satisfaction in hospitals across Australia and to shed light on any areas of support that need improvement.

## Background

Before the 1960s, grief associated with stillbirth was unrecognised in the literature (*Brabin, 2004*). Stillborn babies were taken away and disposed of without any parental involvement and discussion of the loss discouraged (*Lasker & Toedter, 2007*). Over the past 40 years there has been an increase in studies assessing the impact of loss through stillbirth which has created a shift in hospital practices which now encourage bereaved parents to acknowledge and work through their loss (*Leon, 1992*).

Once health professionals realised that attachments were formed before birth during pregnancy, the significance of loss through stillbirth was recognised (*Fetus and Newborn Committee, 2001*; *Leon, 1992*; *Robinson, Baker & Nackerud, 1999*). As early as the 1970s *Yates (1972)* described how mothers of stillborn babies found it helpful to talk about their experiences and how naming the baby was important to them. Additionally, *Lewis (1979)* found that when hospital staff facilitated mourning there was better adjustment to bereavement.

It has been explained by *Leon (1987)* that interactions with the baby and the creation of concrete tokens of remembrance are at the very essence of parental mourning. Not only is the baby to be mourned but also the lost hopes and wishes for a future together, "it is the loss of one who will never be rather the loss of one who once was" (*Leon, 1987*, p. 194).

There is now great value placed on viewing, naming, and holding the baby in order to formulate an identity and validate the pregnancy and death (*Aldridge, 2008*; *Bennett et al., 2005*; *Bonnette & Broom, 2011*; *Callan & Murray, 1989*; *Chance et al., 1983*; *Fetus and Newborn Committee, 2001*; *Hammersley & Drinkwater, 1997*; *Leon, 1987*). It is now commonplace to encourage families to take photographs of the deceased baby as a means of creating an identity for which the baby can be remembered and mourned (*Godel, 2007*). Photographs give the baby social status as a family member and portrays them as a valued individual as well as helping the parents recognise their roles as mother and father, which is often left unrecognised (*Godel, 2007*).

## Individualised support

It has been recommended that health professionals understand the significance of the loss for each unique family as a way to offer appropriate support which should be "open, sensitive, and nondirective, and ultimately tailored..." (*Lafarge, Mitchell & Fox, 2013*, p. 933). Culture, religion, and traditions influence the way people mourn and sensitivity to the variety of beliefs and behaviours may help to facilitate bereavement

(*Bennett et al., 2005*; *Callan & Murray, 1989*; *Chance et al., 1983*; *Chichester, 2005*; *Cowchock et al., 2011*; *Fetus and Newborn Committee, 2001*). It is also important that health professionals do not allow their own personal beliefs to interfere with a family's style of grieving (*Chichester, 2005*; *Flenady et al., 2014*; *Mahan & Calica, 1997*). Understanding the diversity of behaviours, practices, and beliefs held by different cultures will assist with supporting families, although it is advised that assumptions should never be made based on a family's appearance (*Chichester, 2005*). Some families may adhere to cultural norms while others may choose to honour their baby in a unique way (*Chichester, 2005*).

Similarly, it is important to recognise the different coping styles and grief patterns between mothers and fathers (*McGreal, Evans & Burrows, 1997*). In a pilot study assessing sex differences following miscarriage or stillbirth, different rates and forms of grieving were found between mothers and fathers affecting communication and heightening feelings of vulnerability (*McGreal, Evans & Burrows, 1997*). It has been found that fathers feel their grief is less significant in the eyes of hospital staff and therefore often feel overlooked and dissatisfied with treatment (*Bonnette & Broom, 2011*). Mothers have even remarked how their partners were adversely affected by feeling excluded within the hospital post-loss (*Sanchez, 2001*). In a study by *Säflund & Wredling (2006)*, mothers and fathers were found to rate the behaviour of the physician differently, with fathers finding them to be more insensitive than mothers. In the vast majority of research in this area it is common to see only the reactions of mothers documented (*Callan & Murray, 1989*). *Bonnette & Broom (2011)* describe how the recognition and validation of fatherly grief is often overshadowed by the view they are merely supportive partners.

## Nature of support received

In 1982, *Forrest, Standish & Baum (1982)* conducted a randomised trial comparing mothers that received routine hospital care against mothers that received planned support (based on guidelines) and counselling. Planned support and counselling was found to appreciably shorten the duration of distress of bereaved mothers (*Forrest, Standish & Baum, 1982*). However, there is some concern about how protocols and guidelines are handled and applied (*Hutti, 2005*; *Leon, 1992*). It is recommended that care be taken to respect the individual wishes and needs of the family as standardization of bereavement care runs the risk of disrupting a family's unique style of coping (*Bennett et al., 2005*).

Hospital support, or lack thereof, appears to have a significant impact on grief resolution (*Gold, 2007*; *Kirkley-Best & Kellner, 1982*). Parents are able to recount distress caused by negative experiences many years after the event showing how crucial it is that the level of support is meeting the needs and expectations of parents (*Cacciatore & Bushfield, 2007*; *Downe et al., 2012*; *Lafarge, Mitchell & Fox, 2013*). In a recent study by *Crawley, Lomax & Ayers (2013)*, mental health outcomes of bereaved mothers could be predicted by the degree of perceived professional support received. Similarly, *Hammersley & Drinkwater (1997)* found recognition of loss and empathy by others to be powerful factors in alleviating pathological reactions to stillbirth.

With hospital guidelines now in place, there has been a call for more studies to examine the level of support currently offered as well as measuring levels of satisfaction (*Bennett et al., 2005*; *Cacciatore & Bushfield, 2007*; *Callan & Murray, 1989*; *Erlandsson et al., 2011*; *Gold, 2007*; *Lasker & Toedter, 2007*; *Wing et al., 2001*). While some studies indicate that support provided within the hospital setting is satisfactory (*Bennett et al., 2008*; *Conry & Prinsloo, 2008*; *Geerinck-Vercammen & Kanahi, 2003*; *Lafarge, Mitchell & Fox, 2013*), there is evidence that more consistent support is needed (*Cacciatore & Bushfield, 2007*; *Cacciatore, Schnebly & Froen, 2009*; *Conry & Prinsloo, 2008*; *Gold, 2007*; *Lasker & Toedter, 2007*; *Simwaka, de Kok & Chilemba, 2014*). Here, bereaved parents are vital to bring about change by "voicing their stories...and being heard, parents can undo some of the myths that surround stillbirth and focus attention on how things could be done better." (*Scott, 2011*, p. 1388).

Perceived support from health professionals and the opportunity to create memories have been documented as the most important factors in the way parents define their hospital experience (*Conry & Prinsloo, 2008*; *Downe et al., 2012*; *Lafarge, Mitchell & Fox, 2013*). *Forrest, Standish & Baum (1982)* found that almost half of mothers felt that support within the hospital settings could be improved, and where care was judged as satisfactory, flexibility had been mentioned as an important factor.

*Lasker & Toedter (2007)* completed one of the largest longitudinal studies assessing parents' satisfaction with hospital care in the United States. They found that parents who received support were more satisfied than parents that did not, although having more support did not lead to greater levels of overall satisfaction. They suggest this is because quality of perceived support is also a determinant of satisfaction and grief alleviation. *Harper & Wisian (1994)* have also documented a significant positive relationship between satisfaction and the use of most recommended interventions. Variation in the support offered by different health care professionals has also been reported, with doctors often being described as not meeting the emotional needs of parents (*Cacciatore, Schnebly & Froen, 2009*; *Erlandsson et al., 2011*; *Gold, 2007*). In Australia, *Brabin (2004)* discusses how the Stillbirth and Neonatal Death Support (SANDS) charity has reported a shift over the last 20 years with fewer complaints regarding hospital support and an increase with the satisfaction of care. In the UK, parents that had worked collaboratively with health professionals were able to create bereavement support that "respects parents' needs, acknowledging that stillbirths matter and that the quality of care grieving parents receive can have a lifelong effect..." (*Scott, 2011*, p. 1388).

Very little research of a quantitative nature has been employed to assess parents' level of perceived support and whether the support received is satisfactory. The majority of studies have also used very small sample sizes and have evaluated only one hospital service. Furthermore, only a limited number of studies have assessed the experiences of both mothers and fathers. No studies have examined the support offered in Australian hospitals and whether parents of stillborn babies have found such support satisfactory (perceived support). Additionally, limited studies have examined differences in perceived support and

satisfaction following the introduction of the PSANZ guidelines. The current study aims to address these shortcomings.

Although designed to be non-prescriptive, the PSANZ guidelines have been used by the current study as a benchmark to examine the support offered to parents of stillborn babies in Australian hospitals. These guidelines are designed to provide knowledge to Australian and New Zealand health professionals by describing generally recommended practice and to enhance the quality of bereavement care (*Flenady et al., 2009*). Following the recommendations of the National Health and Medical Research Council, the guidelines were developed by searching for existing guidelines and completing a comprehensive literature review (*Flenady et al., 2009*). Support offered is used in the current study as the collective term to include the hospital practices and interventions (guidelines) relevant to stillbirth under the following categories: respect, information, autopsy, birth options, hospital stay, creating memories, and aftercare. By examining levels of perceived support and satisfaction it is hoped that the research will shed light on any areas that need improvement.

The present study aims:

1. To assess the level of perceived support parents are receiving (as determined by the PSANZ guidelines) and the extent of parent satisfaction.
2. To identify if the introduction of the PSANZ guidelines has significantly increased levels of perceived support and parent satisfaction and whether there is a negative correlation between years since birth and support and satisfaction.
3. To examine the extent to which perceived support is positively correlated with parent satisfaction and to explore which areas of support parents perceive to be most influential to their grief.
4. To determine if fathers are receiving a lower level of perceived support and are less satisfied than mothers.

## METHOD

### Participants

Participants were at least 18 years old and were parents of stillborn babies born in an Australian hospital. A stillbirth occurs when a baby dies at 20 weeks or more gestation or with a birth weight of at least 400 grams. Participants were asked about gestation but not birth weight. Two participants, who gave birth on or before 12 weeks gestation, where it was reasonable to assume they had a miscarriage as opposed to a stillbirth, were excluded. One-hundred and forty-eight participants were excluded as they answered very few survey questions while one participant was excluded because she gave birth outside of Australia.

The sample consisted of 181 women (95.8%), six men (3.2%), and two (1.1%) who did not indicate if they were a woman or a man. Age ranged from 18 to 65 years ($M = 34.9, \text{SD} = 7.6$). The majority of participants were married/de facto (84.1%) and generally well educated with 41.8% having completed a university degree or higher.

Years since birth ranged from zero to 40 years ($M = 5.0, \text{SD} = 6.4$) and time of birth ranged from 16 to 42 weeks gestation ($M = 30.5, \text{SD} = 7.5$). Location of birth covered

all states and territories across Australia: New South Wales (32.3%), Victoria (28.6%), Queensland (18.0%), Western Australia (9.0%), South Australia (5.8%), Australian Capital Territory (2.6%), Tasmania (2.6%), and Northern Territory (1.1%). According to the data collected by Australia's Mothers and Babies 2011 (*Li et al., 2013*), this is a fairly accurate cross-section of the population. Participants in the study gave birth in 96 different hospitals across Australia with 82.5% being public and 17.5% being private. One-hundred and thirty-three participants were admitted as public patients (70.4%), 54 were admitted as private patients (28.6%), and two did not report patient admission status (1.1%). A total of 174 (92.1%), of the pregnancies were single, with 14 (7.4%), being a multiple pregnancy, and one (0.5%), participant failing to report. Of the multiple pregnancies, nine participants reported a loss of one baby, while five reported a loss of more than one.

## Procedure

The study was conducted with the approval of the Human Research Ethics Committee of the University of New England, Australia, approval number HE14-149. Participants were recruited online via Facebook where approval was sought from page administrators for a link to the Stillbirth Support Survey to be posted on the page. Pages were selected by searching key words such as "stillbirth," "stillborn," and "pregnancy loss" and any linked and suggested pages on the chosen pages were also explored. The pages included closed groups, open groups, non-profit organisations, communities, charity organisations, and public figures. Of the 27 pages approached, seven agreed to participate. The link was voluntarily shared on other pages, recommended to specific individuals, and displayed on one counselling website.

   Potential participants who followed the link were forwarded to the Stillbirth Support Survey powered by Qualtrics software (*Qualtrics, 2014*). An *information* sheet was initially provided followed by an online implied *consent* form where potential participants could choose to proceed or exit the study. If participants chose to proceed, then they were given survey instructions before going on to complete the Stillbirth Support Survey. If a participant had experienced more than one stillbirth from separate pregnancies questions were to be answered based on the most recent experience. If a particular question was not relevant or if participants could not recall if something did or did not happen, they were advised to leave the item blank. To reduce the risk of emotional trauma, participants were reminded that they were able to withdraw from the survey at any stage and information for bereavement counselling was made available at the beginning and end of the survey.

## Materials

Participants volunteered to complete the Stillbirth Support Survey designed by the researcher. The survey consisted of 12 demographic and topic-specific questions followed by 50 questions assessing perceived support as per the PSANZ guidelines. Here, perceived support was measured under the following categories: respect, information, autopsy, birth options, hospital stay, creating memories, and aftercare. Participants were presented with a statement and asked to answer on a 5-point scale (1 = *strongly disagree* to 5 = *strongly agree*). Example items include: "My cultural/religious beliefs, traditions and practices

were respected by hospital staff," "The information I received was delivered in a sensitive manner," and "I was provided with an opportunity to bathe my baby/babies." Participants were asked how satisfied they were with the overall level of perceived support on a 7-point scale (1 = *very dissatisfied* to 7 = *very satisfied*), and then asked which area(s) of support they felt were most influential to their grief process (respect, information, autopsy, birth options, hospital stay, creating memories, aftercare, and other). The perceived support and satisfaction measures where collapsed (i.e. "agree" and "strongly agree" were merged into one) for Aim 1 to assist the interpretation of results.

No reportable measures of reliability and validity are available for the Stillbirth Support Survey. Questions were based directly on the published PSANZ guidelines which are intended as generally recommended practice for Australia and New Zealand (*Flenady et al., 2009*).

## Statistical analysis

Pearson's correlation coefficient and independent *t*-tests were used to analyse the data. Assumptions of normality, independence, and homogeneity of variance were met for the "satisfaction" and "perceived support" variables. Due to positive skewness and kurtosis of the "years since birth" variable, a $log + 1$ transformation was conducted as per *Field*'s (*2005*) recommendation. There was little difference between results for raw and transformed data thus raw data results were retained for ease of interpretation.

## RESULTS

### Perceived support (Aim 1)

Descriptive statistics were used to assess levels of perceived support (as determined by the PSANZ guidelines). Participants who indicated either *agree* or *strongly agree* on support items were combined to determine the extent to which guidelines are being followed in Australian hospitals. Table 1 provides a summary these findings.

### Parent satisfaction (Aim 1)

Descriptive statistics were used to assess the amount of satisfaction among parents of stillborn babies. In total, of the 189 participants, 64.0% reported satisfaction with their overall level of perceived hospital support. There were 30.2% reports of dissatisfaction and 5.8% of participants remained neutral.

### The effect of the PSANZ guidelines on support and satisfaction (Aim 2)

The PSANZ guidelines were published in 2009 so the participants were split into two groups based on whether participants gave birth before or after the release of the material. Mean scores for perceived support and satisfaction are provided in Table 2.

On average, parents who gave birth after 2009 reported greater perceived support ($M = 3.51, \mathrm{SE} = 0.07$), than parents who gave birth before 2009 ($M = 3.12, \mathrm{SE} = 0.12$), $t(186) = 3.12, p < .001$ with a small to medium sized effect $r = .22$. Similarly, parents who gave birth after 2009 also reported higher satisfaction ($M = 5.11, \mathrm{SE} = 0.17$) than parents

**Table 1  Percentage of Participants that 'agree' or 'strongly agree' on the Different Support Items as per the PSANZ Guideline.**

| Measure | % |
|---|---|
| **Respect** | |
| Respect to baby/babies | 76.7 |
| Respect to cultural/religious beliefs | 71.0 |
| Grief validated by hospital staff | 74.0 |
| Supported to reach own decisions | 66.5 |
| **Information provision by hospital staff** | |
| To both parents | 73.0 |
| Sensitively | 75.2 |
| Honestly | 75.7 |
| Clearly | 67.7 |
| Appropriate terminology | 78.6 |
| In a quiet private place | 81.5 |
| Appropriate time | 58.2 |
| Written fact sheets | 66.5 |
| **Adequate time to** | |
| Consider information | 55.5 |
| Ask questions | 63.5 |
| Grieve silently | 68.8 |
| **Hospital staff** | |
| Made sure that parents understood the information provided | 69.2 |
| Spoke of the baby/babies in sensitive terms | 75.7 |
| **Autopsy** | |
| Verbal and written options for post-mortem | 63.0 |
| Informed that results might take months to return and that noting adverse may actually be found | 66.3 |
| Received information in a quite private place | 58.6 |
| Felt comfortable with the person delivering the information who could competently answer questions | 52.4 |
| Knew the person taking their baby/babies to autopsy | 11.1 |
| Given opportunity to meet the pathologist and assured their baby/babies would be treated with respect | 6.3 |
| Given the option to see/hold their baby/babies after the autopsy | 28.3 |
| **Birth Options** | |
| Given relevant information regarding delivery | 31.7 |
| Provided with a choice to remain in hospital or return home prior to delivery | 45.4 |
| Provided with information on the benefits and consequences of each type of delivery (natural vs. caesarean section) | 40.5 |
| Offered a choice in birth options | 31.4 |
| **Hospital environment** | |
| Asked to select a ward | 13.1 |
| Provided with a private room | 93.1 |
| Away from the busiest part of the ward | 55.0 |
| Symbol placed on their door | 39.9 |
| Continuity of care | 60.3 |
| Time available with their baby/babies | 85.7 |
| Staff member available to collect/return baby/babies as desired | 57.1 |
Table 1 (*continued*)

| Measure | % |
|---|---|
| Informed that there was no urgency to leave the hospital | 59.8 |
| Social worker provide support, counselling and information | 57.2 |
| **Opportunities to create memories** | |
| Informed participants the length of time they could spend with their baby/babies | 42.3 |
| Gave option to stay in hospital or take their baby/babies home | 17.0 |
| Informed what to expect in terms of the appearance of their baby/babies | 55.7 |
| Made participants aware that there was no urgency to arrange a funeral | 37.1 |
| Informed participants that a baptism/blessing could be arranged if desired | 42.5 |
| Bathe their baby/babies | 35.1 |
| Had staff provide hand and footprints/ID bracelet/photographs/cot cards | 86.7 |
| Had staff make suggestions of the creation of memories | 57.7 |
| **Perceived after care support** | |
| Informed about milk production and was given the option of a lactation consultant | 42.0 |
| Made aware of post-pregnancy changes and the need for a post-birth check-up | 82.3 |
| Received information and referrals to other relevant health professionals | 53.5 |
| Informed of the legal requirement to arrange a funeral and given options for funeral arrangements | 76.5 |
| Told what to expect in terms of the grief process | 37.2 |

**Table 2** Mean scores on the stillbirth support survey.

| Measure | Births from 1974–2008 (n = 56) | | Births from 2009–2014 (n = 132) | |
|---|---|---|---|---|
| | **M** | **SD** | **M** | **SD** |
| Respect (4) | 3.67 | 1.08 | 3.98 | 1.01 |
| Information (13) | 3.39 | 1.04 | 3.83 | 0.93 |
| Autopsy (7) | 2.81 | 0.83 | 3.19 | 0.80 |
| Birth options (4) | 2.68 | 1.07 | 3.03 | 0.96 |
| Hospital stay (9) | 3.10 | 0.93 | 3.60 | 0.85 |
| Creating memories (8) | 2.84 | 0.92 | 3.18 | 0.83 |
| Aftercare (5) | 3.10 | 1.01 | 3.48 | 0.96 |
| Total perceived support score (50) | 3.12 | 0.87 | 3.51 | 0.75 |
| Satisfaction (1) | 4.09 | 2.17 | 5.11 | 1.97 |

**Notes.**

Perceived support scores could range from 1 (*strongly disagree*) to 5 (*strongly agree*), and satisfaction scores ranged from 1 (*very dissatisfied*) to 7 (*very satisfied*). Number of items for each measure is displayed in parentheses.

who gave birth before 2009 ($M = 4.09$, SE $= 0.29$), $t(186) = 3.16, p < .001$ with a small to medium sized effect $r = .23$. A description of satisfaction frequencies and percentages between groups is provided in Table 3.

## Perceived support and satisfaction over time (Aim 2)

Pearson's correlation coefficient was used to determine whether there was a negative relationship between perceived support, satisfaction, and years since birth. Perceived support was negatively correlated to years since birth, with a coefficient of $r = -.33$,
**Table 3 Satisfaction frequencies and percentages in relation to guidelines.**

| Satisfaction | Births from 1974–2008 ($n = 56$) | | Births from 2009–2014 ($n = 132$) | |
|---|---|---|---|---|
| | Frequency | % | Frequency | % |
| Very dissatisfied | 8 | 14.3 | 11 | 8.3 |
| Dissatisfied | 10 | 17.9 | 10 | 7.6 |
| Somewhat dissatisfied | 8 | 14.3 | 10 | 7.6 |
| Neutral | 4 | 7.1 | 7 | 5.3 |
| Somewhat satisfied | 5 | 8.9 | 19 | 14.4 |
| Satisfied | 11 | 19.6 | 34 | 25.8 |
| Very satisfied | 10 | 17.9 | 41 | 31.1 |

**Table 4 Categories parents perceive to be most influential to their grief ($n = 189$).**

| Measure | Frequency | % |
|---|---|---|
| Creating memories | 142 | 75.1 |
| Respect | 125 | 66.1 |
| Hospital stay | 110 | 58.2 |
| Aftercare | 86 | 45.5 |
| Information | 84 | 44.4 |
| Autopsy | 36 | 19.0 |
| Birth options | 34 | 18.0 |
| Other | 22 | 11.6 |

which was also significant at $p < .01$. A negative relationship was also documented between satisfaction and years since birth ($r = -.31, p < .01$).

## Relationship between perceived support and satisfaction and influences on grief (Aim 3)

Pearson's correlation coefficient was also used to determine if there was a positive relationship between perceived support and satisfaction. A significant relationship was found, $r = .89, p < .01$.

Descriptive statistics were used to explore which areas of perceived support parents perceived to be most influential to their grief, see Table 4. Over half of the participants reported creating memories, respect, and hospital stay to be the most influential.

## Differences between mothers and fathers (Aim 4)

The sample did not enable a meaningful comparison of mothers and fathers given the small sample of fathers.

## DISCUSSION

The current study aimed to assess the level of perceived support parents are receiving (as determined by the PSANZ guidelines) and the extent of parent satisfaction. Over

the last 40 years, Australian hospitals have provided some relevant support to bereaved parents of stillborn babies; however, much improvement is needed. While some guidelines are being demonstrated in up to 93.1% of cases, others are rarely displayed (as low as 6.3%). On average, guidelines are being implemented just over 55% of the time. Overall, areas of perceived support that are the strongest include respect, information, hospital stay, and after care, with birth options, creating memories, and autopsy being the lowest scoring categories on average. It is evident that there is considerable variance in terms of support with some bereaved parents being offered specific interventions and practices while others are not. For instance, some parents are missing out on important information, opportunities with their deceased baby/babies, and options regarding their care. Overall, of the 189 participants that gave birth between 1974 and 2014, 64% reported satisfaction with the overall level of perceived support.

As expected, a significant increase in both perceived support and parent satisfaction was found since the publication of the PSANZ guidelines in 2009. Further support for guidelines has been found in the UK where the duration of distress was noticeably shortened with the introduction of guidelines and counselling (*Forrest, Standish & Baum, 1982*).

Of the participants who gave birth prior to 2009, 46.5% reported some degree of dissatisfaction compared to only 23.5% of participants who gave birth after 2009. Although this is a marked decrease, this means that almost one in four bereaved parents leave the hospital not only mourning the loss of their child, but disappointed with the support provided to them at one of the most difficult times of their lives. These findings are not dissimilar to other recent studies which have reports of support still being inadequate, inconsistent, and unsatisfactory for some bereaved parents (*Bonnette & Broom, 2011*; *Cacciatore & Bushfield, 2007*; *Conry & Prinsloo, 2008*; *Downe et al., 2012*; *Erlandsson et al., 2011*; *Lafarge, Mitchell & Fox, 2013*; *Simwaka, de Kok & Chilemba, 2014*). Distress caused by negative experiences can be long-lasting which indicates how crucial it is that all bereaved parents are adequately supported (*Cacciatore & Bushfield, 2007*; *Downe et al., 2012*; *Kirkley-Best & Kellner, 1982*; *Lafarge, Mitchell & Fox, 2013*).

As hypothesised, perceived support and satisfaction were negatively correlated with years since birth. This suggests that there has been some improvement of support, and higher rates of satisfaction, over time. These results confirm reports that there have been far fewer complaints and greater satisfaction among parents of stillborn infants in Australia over the last 20 years (*Brabin, 2004*). Midwives have embraced training opportunities (*Brabin, 2004*), so it is hoped that as health professionals continue to become more experienced and knowledgeable, improvements will continue to be seen.

As hypothesised, a significant and strong positive relationship between perceived support and parent satisfaction was documented. It seems likely that as health professionals adhere to the guidelines provided by PSANZ, greater satisfaction will be seen among bereaved parents. However, a direct causal relationship cannot be determined by the present study. In the United States, *Harper & Wisian (1994)* also documented support for guidelines with satisfaction increasing with most recommended practices. Although

the guidelines are designed to be non-prescriptive, the results suggest that following the recommendations by PSANZ may have a positive impact on bereaved parents.

The study also sought to explore which areas of support parents perceive to be most influential to their grief. Three quarters of participants deemed creating memories (time with baby/babies, mementoes, and baptism/blessing) to be the most influential area of support. Around half of the participants also reported respect (for baby, parents, and cultural/religious beliefs), hospital stay (environment, support of staff), after care (maternal changes, support services, referrals, and expectations of grief), and information (timing, delivery, mode, and terminology) to be most influential to their grief. These results support previous research that claims the creation of quality memories is highly valued among bereaved parents (*Bonnette & Broom, 2011*; *Conry & Prinsloo, 2008*; *Crawley, Lomax & Ayers, 2013*; *Godel, 2007*; *Lafarge, Mitchell & Fox, 2013*; *Lasker & Toedter, 2007*). Additionally, the quality of memories has been found to dictate whether the experience was viewed as either positive or negative (*Downe et al., 2012*). The creation of memories further assists in creating an identity for the deceased as well as validating the pregnancy and the loss (*Aldridge, 2008*; *Bennett et al., 2005*; *Bonnette & Broom, 2011*; *Callan & Murray, 1989*; *Chance et al., 1983*; *Fetus and Newborn Committee, 2001*; *Hammersley & Drinkwater, 1997*; *Leon, 1987*). Furthermore, sharing memories has been associated with fewer symptoms of post-traumatic stress disorder (*Crawley, Lomax & Ayers, 2013*).

With such a high value placed on creating memories by parents and the associated benefits to mental health (*Crawley, Lomax & Ayers, 2013*; *Rådestad et al., 1996*), it is a concern that this category was one of the lowest scoring in terms of perceived support. This highlights that considerable changes need to be made by health professionals to ensure that support is consistent, relevant, and meeting the needs and expectations of bereaved parents.

## Limitations

The sample may have been skewed given that it was predominantly younger, female, married, and well-educated. All participants also had internet access. Given that stillbirth affects a very small percentage of the population, there is difficulty in the recruiting of participants, however, future research should seek to engage a more diverse and representative sample. The use of some online support groups used for recruitment may have introduced bias. Participants were asked to comment retrospectively so there is also the potential for recall bias in the rating of experiences. However, it appears that although parents were informed to leave items blank if they could not remember or were unsure, many were still able to recount experiences from many years ago.

The recruitment process may have biased the diversity of the sample. Although attempts were made to include male participants by seeking support groups for fathers, the majority of pages that accepted participation were support groups for mothers Additionally, it has been documented that while women are more expressive, men tend to exert more control over their emotions and are expected to keep their feelings to themselves (*Stinson et al., 1992*; *Wing et al., 2001*). Such attitudes may have impacted self-inclusion rates among

fathers in the current study. Furthermore, it has been reported that finding a sense of belonging is the foremost coping strategy used by mothers of stillborn babies (*McGreal, Evans & Burrows, 1997*). It may be that more mothers than fathers sought out the selected pages on Facebook (where the survey was advertised) as a way of connecting with others and dealing with their grief. An alternate method of recruitment is advised for future studies aiming to include male participants.

It is possible that participants may have previously been satisfied with hospital support until completing the Stillbirth Support Survey. Participants may have been unaware of specific interventions and practices that could have been offered which in turn may have lowered their original perceptions of the support they received. Future research in this area could evaluate satisfaction at the beginning and end of the questionnaire to determine if perceptions are altered.

Furthermore, the study reports on experiences up to 40 years ago where it is recognised that parental and societal expectations would have differed from today. However, even in the early 1970s, research was highlighting the benefits of seeing the baby, talking about the loss, naming the baby, and having a private room following the birth (*Yates, 1972*). In an attempt to report whether guidelines are making a difference to perceived support and satisfaction, it was important to gather data on experiences before and after the PSANZ guidelines were released. It was also important to represent the mothers and fathers who gave birth to a stillborn baby many years ago and had a traumatic experience due to the lack of perceived support.

## CONCLUSION

Although perceived support for bereaved parents in Australia has improved since the introduction of the PSANZ guidelines, it is clear that hospitals are not implementing them fully. However, it is unclear why this is happening. It may be that there is still knowledge lacking among health professionals, or perhaps it is due to insufficient training and review processes.

Parents in the current study reported higher levels of satisfaction when hospitals were following the recommendations. Creating memories was regarded as the most influential area of support by most participants, however, it was one of the categories most lacking support. It is recommended that health professionals within the hospital environment review guidelines that are in place and seek feedback from parents as to how support can be improved.

### Funding
The authors declare there was no funding for this work.

### Competing Interests
The authors declare there are no competing interests.

## Author Contributions

- Melanie L. Basile conceived and designed the experiments, performed the experiments, analyzed the data, contributed reagents/materials/analysis tools, wrote the paper, prepared figures and/or tables, reviewed drafts of the paper.
- Einar B. Thorsteinsson conceived and designed the experiments, contributed reagents/materials/analysis tools, wrote the paper, prepared figures and/or tables, reviewed drafts of the paper.

## Human Ethics

The following information was supplied relating to ethical approvals (i.e., approving body and any reference numbers):

This project was approved by the Human Research Ethics Committee of the University of New England, Australia. Approval No. HE14-149.

## Data Deposition

The following information was supplied regarding the deposition of related data:

The corresponding author will take requests from academics and researchers that would like to view/analyse the data further. Please contact Einar Thorsteinsson, Ph.D. via email: ethorste@une.edu.au or einarbt@gmail.com.

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
