# Peer review of "Parents' evaluation of support in Australian hospitals following stillbirth"

_PeerJ, doi:10.7717/peerj.1049_

## Round 0.1 · original submission · Major Revisions

Please take careful note of all issues raised by reviewers and provide a point by point response indicating how the issue was addressed and where in the revised manuscript these changes can be found.

·

Basic reporting

- The introduction is not focused enough and needs to be more concise.
- Perceived support (as opposed to actual support) and satisfaction with care needs to be introduced and adequately defined.
- I would focus the introduction on mothers, given that your sample does not allow you to draw any conclusions about fathers.
- The theoretical and societal background of the PSANZ guidelines should be discussed. How do these guidelines compare with guidelines in other countries?
- P. 4, line 14: I would either add other theoretical perspectives or take the first sentence out.
- The results need to be summarised and reported more concisely. I would suggest to put everything into a table and to only mentioned those findings that are most interesting from a theoretical or clinical perspective. Currently, important findings are buried under more trivial findings. I would encourage the authors to categorise the findings according to concepts or themes.
- P.14, line 14: please reword, as 'etcetera' sounds slightly disrespectful.
- P.13, line 18: were participants only asked whether they saw or held their stillborn after the autopsy took place? What about the contact with the stillborn after the birth?
- Throughout the paper, it is very important to replace "support" with "perceived support" and to be aware that this is what you measured.
- P.3, line 22: replace "bereavement" with "bereaved"
- P.6, line 1: year of publication is missing
- P. 7, line 2: take "that" out
- P. 7, line 4: insert "a" call
- P. 7, line 22: insert ", although"
- P. 8, line 5: replace "organisations have" with "Charity has"
- P. 9, line 18: "who did not attempt" - what do you mean?
- P. 9, line 21: replace "sex" with "gender"
- P. 11, line 1: "the support and satisfaction measures were collapsed" - what do you mean?
- P. 12, 2nd para: take whole paragraph out. There is no need to report a power analysis, given that you only had a subsample of 6 fathers.
- P. 13, line 12: "and that nothing adverse may be reported" - needs to be reworded
- P. 13, line 14: replace "and" with "who"
- P. 14, line 21: replace "journey" with "process"
- P. 15, line 18: you can delete the explanation of effect sizes
- throughout the manuscript (including the abstract), please make sure you talk about "perceived support" rather than "support".

Experimental design

- An important flaw of the design is the range of years since stillbirth (up to 40 years), which would significantly bias any recall. I find it problematic to compare perceived support and satisfaction before and after the guidelines were introduced (2009), as parental and societal expectations would have been very different 40 years ago compared with today. This needs to be discussed in more detail under study limitations.
- Inclusion and exclusion criteria are missing.

Validity of the findings

- Given that the authors only had 6 fathers in your sample, their fourth research questions cannot be addressed at all. You therefore need to take the first para on p. 17 as well as the second para on p. 20 out. Just say briefly that your sample did not allow you to examine this research question and do not attempt to discuss this question despite the lack of fathers in your sample.
- The authors should address the lack of fathers in their sample in their discussion. Was this due to a biased recruitment strategy? Or was the design of the questionnaire less appealing/relevant for fathers?

Additional comments

This is an interesting study that addresses an important question of maternal perceived support and satisfaction with care following a stillbirth in Australian hospitals. However, some major revisions need to take place before the manuscript is publishable.

Reviewer 2 ·

Basic reporting

This paper reports on an important but under-researched topic – the quality of care for parents who have experienced stillbirth. The study reports findings that suggest improvements have occurred over time together with tentative evidence that these improvements have followed the development of guidelines for care. While this is encouraging, the study also clearly indicates that suboptimal care is still not uncommon.

The paper generally is clearly written although there are instances where wording or expression might be improved.

The structure of the paper seems appropriate. The material under the heading “Accessing Support” seems less about parents’ accessing support than about the nature of support received and the heading title could better reflect this.

I would expect to see reference to at least one of the papers from the 2011 Lancet Stillbirth Series under Scope of the Problem and/or Background sections.
Under Background P4, Line 2 – specific mention is made to the nursing literature but this is true of the literature more broadly.

Specific reference is made to the psychoanalytic perspective. It is not clear why just this one theoretical perspective is mentioned here. Other theories also account for the importance of interaction with the baby (e.g., continuing bonds theory) and perhaps some more overarching statement could be made in stressing the importance of interactions for memory making.

In some places, single older references are used to support some now quite widely documented findings. For example, P5, Line 22 – McGreal et al, 1997, a pilot study on mothers and fathers coping with grief. Similarly, P5, Lines 15-21 and the single Chichester reference. Hospital chaplains are one important source of information to assist with the provision of culturally appropriate support but there are others, including social workers, Aboriginal & Torres Strait Islander liaison services etc. Cacciatore’s work around parent-centred care and provision of culturally appropriate care would also be relevant.
P.11, Line 1, “ease with” should be replaced with “aid” or “assist”
P.12 Line 21 – suggest replacing “rate” with “extent” as rate as a particular meaning.
P.13, Line 24 – delete the word “Various”
P.14, Line 14 – replace “etcetera” with “similar mementoes”
P.18, Line 6-8 – suggest rewording the sentence so that it is more informative for readers – e.g. found in hospital settings in UK (rather than found in Oxford).
The abstract should include the number of mothers and fathers to show that the study predominantly includes mothers as well as the recruitment methods.

Experimental design

I would like to see information about how participants were recruited placed earlier in the methods section.
Is it possible to provide any further detail about the sources of recruitment for those who participated?
The survey appears to have been conducted rigorously within the constraints of the study design. Although a survey of parents recruited from those who have contacted online organisations raises particular issues in terms of sample bias, it is important to acknowledge the difficulty of recruiting participants for studies of this type (perhaps the authors could mention this in the Limitations section).

Validity of the findings

The tentative nature of the findings does need to be emphasised given the study design and sample bias.
It is not clear how many hospitals had implemented the guidelines or how comprehensively. This broader issue of guideline implementation could also be mentioned in conclusions.

---

## Round 0.2 · accepted · Accept

The authors have fully addressed all issues previously raised by reviewers.

·

Basic reporting

No comments

Experimental design

No comments

Validity of the findings

No comments

Additional comments

Many thanks to the authors for responding with great care to all of my comments. I am satisfied that all of my queries have been addressed and believe that the manuscript has significantly improved.

Reviewer 2 ·

Basic reporting

No further comments as this is a re-submitted article

Experimental design

No further comments as this is a re-submitted article

Validity of the findings

No further comments as this is a re-submitted article

Additional comments

I am satisfied that the authors have addressed the comments I raised in my initial review.